

**Unveiling Hydrological Dynamics in Data-Scarce Regions:**
**A Comprehensive Integrated Approach**
[1]Ayenew D. Ayalew, [1] Paul D. Wagner, [2] Dejene Sahlu, and [1] Nicola Fohrer
[1] Department of Hydrology and Water Resources Management, Christian-Albrechts-
University, Kiel, Germany
[2] Institute of Disaster Risk Management and Food Security Studies, Bahir Dar University
Email: ayalew@hydrology.uni-kiel.de
**Abstract:** The hydrological system of Rift Valley Lakes in Ethiopia has recently experienced changes
since the past two decades. Potential causes for these changes include anthropogenic, hydro-climatic
and geological factors. The main objective of this study was to utilize an integrated methodology to
gain a comprehensive understanding of the hydrological systems and potential driving factors within a
complex and data-scarce region. To this end, we integrated a hydrologic model, change point analysis,
indicators of hydrological alteration (IHA), and bathymetry survey to investigate hydrological dynamics
and potential causes. A hydrologic model (SWAT+) was parameterized for the gauged watersheds and
extended to the ungauged watersheds using multisite regionalization techniques. The SWAT+ model
performed very good to satisfactory for daily streamflow in all watersheds with respect to the objective
functions, Kling–Gupta efficiency (KGE), the Nash–Sutcliffe efficiency (NSE), Percent bias (PBIAS).
The findings reveal notable changes of lake inflows and lake levels over the past two decades. Chamo
Lake experienced an increase in area by 11.86 km², in depth by 4.4 m, and in volume by 7.8 x 108 m³.
In contrast, Lake Abijata witnessed an extraordinary 68% decrease in area and a depth decrease of 1.6
m. During the impact period, the mean annual rainfall experienced a decrease of 6.5% and 2.7% over
the Abijata Lake and the Chamo Lake, respectively. Actual evapotranspiration decreased by 2.9% in
Abijata Lake but increased by up to 0.5% in Chamo Lake. Surface inflow to Abijata Lake decreased by
12.5%, while Lake Chamo experienced an 80.5% increase in surface inflow. Sediment depth in Chamo
Lake also increased by 0.6 m. The results highlight that the changing hydrological regime in Chamo
Lake is driven by increased surface runoff and sediment intrusion associated with anthropogenic
influences. The hydrological regime of Abijata Lake is affected by water abstraction from feeding rivers
and lakes for industrial and irrigation purposes. This integrated methodology provides a holistic
understanding of complex data scarce hydrological systems and potential driving factors in the Rift
Valley Lakes in Ethiopia, which could have global applicability.
**Keywords Hydrological change; IHA; Bathymetry; Lake Chamo; Lake Abijata; SWAT+; Rift**
**Valley; Lake level.**



# 1 Introduction

Continuous anthropogenic and natural activities are adversely altering the water resources from local to global scales through land use change, water use and climate change (Flörke et al., 2018, Sivapalan et al., 2003). Changes of lake levels influence ecosystem services worldwide (Gownaris et al., 2015, Kolding and van Zwieten, 2012). In Eastern Africa, the Rift Valley Lakes are supporting eco-regions of great biodiversity, considered amongst the global 200 freshwater eco-regions of the world (O'Brien et al., 2018, Olson and Dinerstein, 2002). In this region, water resources play a crucial role for economic development through use of water for domestic supply, irrigation and industry, fishery, recreation and tourism (Cowx and Ogutu-Owhayo, 2019, Minale, 2020, White et al., 2002).

The Ethiopian Rift Valley Lake basin is known as the Lakes District, containing a chain of eight main lakes situated at the Rift floor. The total surface area of open waters, including wetlands of the Rift valley was 3413 km2 in 2019 which is 46% of total area of open waters resources of Ethiopia (Ayalew et al., 2022). Most of these lakes are highly productive, contain indigenous populations of edible fish, and support a variety of aquatic and terrestrial wildlife (Tudorancea and Taylor, 2002). Currently, the hydrology of the Rift Valley Lakes is changing, i.e. water levels, areal extent, and volumes are altered, affecting ecosystem services and the communities. The situation has been aggravated  recently due to increasing population density, excessive water abstraction and catchment land use changes (Billi and Caparrini, 2006, Legesse and Ayenew, 2006). Because of high water abstraction for irrigation and industry, as well as climate change, the open water surface area has decreased by 2.1% in three decades (Ayalew et al., 2022). The problem is getting worse since water abstractions is often carried out without a basic understanding of the complex hydrogeological system, and the fragile nature of the Rift ecosystem (Zinabu et al., 2002, Ayenew, 2002). The rapidly growing irrigation and improper use of water resources has resulted in notable shirking of waterbodies in the central Rift Valley Lakes as e.g. Lake Abiyata has lost 68% of its surface area within the last four decades (Ayalew et al., 2022). The feeding rivers and some lakes are being used to meet various human needs, playing an increasingly central role in the lives of millions of people. The basin has become economically significant because of the development of flower and horticultural production, soda ash factory, tourism, and other human activities around the shores of the lakes. The basin's future economic development will be linked to water, but the water resources are changing due to climate change and human induced factors. Despite the basin's significance as a vital



component of the country's economy and ecological balance, the major challenge lies in the
limited availability of accurate hydrological data. Previous hydrological studies conducted in
the Rift Valley Lakes have predominantly focused on analyzing the impact of single drivers
and have failed to provide a quantitative assessment of the respective contributions of land use
and climate change on streamflow. For instance, some studies have primarily investigated the
effects of high water abstraction for irrigation and industries (Kebede et al., 1994, Zinabu and
Elias, 1989, Legesse and Ayenew, 2006, Ayenew, 2002) , environmental degradation (Ayalew
et al., 2004, Meshesha et al., 2012, Ayenew, 2004), volcano-tectonics, and sedimentation (Le
Turdu et al., 1999, Street and Grove, 1979), bathymetry analysis (Awulachew, 1999) as well
as the occurrence of frequent earthquakes and faults  (Ayalew et al., 2004, Ayenew, 2002,
Belay, 2009). However, the hydrological system of the Rift Valley is very complex and
processes within a basin are driven by the interplay of climate, LULC, topography, soil and
human activities. Evaluating hydrological change typically involves assessing changes in the
flow regime and water balance of a river, a lake or a reservoir. Therefore, to understand the
hydrological system and driving forces that control the hydrological processes requires an
integration approach. The methods used to evaluate hydrological change depend on the specific
goals of the evaluation and the data available. Flow regime analysis, trend analysis, time series
analysis, and hydrological modelling are common methods used for hydrological change
analysis (Wagener and Montanari, 2011, Hargreaves and Samani, 1985, Jain and Singh, 2019);
and in regions with limited available hydrological data, the application of regionalization
techniques is a commonly employed approach(Viglione et al., 2013, Pool et al., 2021, Pagliero
et al., 2019).
The study aims to understand the changing hydrological systems and driving factors by
integrating hydrological modelling, indicators of hydrological alteration (IHA), change
point/break analysis, and bathymetry survey analysis. We hypothesize that the changes in the
hydrological regimes are associated with changes in high water abstraction, climate change and
land use.

## Materials and methods

### 2.1.   Study Area Discerption

The Ethiopian Rift Valley Lakes Basin is part of the Great Rift Valley, which is a 4,000 km
long fault line that stretches from the Red Sea to Mozambique's and Zambia's Valley. It



transects Ethiopia, Kenya, Uganda, Rwanda, Burundi, Zambia, Tanzania, Malawi and
Mozambique (Gregory, 2018). The Ethiopian part of the Rift Valley Lakes basin is located
between 36° and 40°E and 4° and 9°N. It extends from the Afar depression southwards to
Kenya across the broad basins of Abijata-Ziway, Abaya-Chamo, and Segen (Fig. 1). It is one
of the most important basins in Ethiopia and occupies an area of 55,050 km². It characterized
by diverse landscapes features and climate conditions. It has a complex hydrological system
and encompasses numerous lakes, springs, wetlands and rivers. Given its distinctive attributes,
it lends itself well to the development of a globally applicable methodology.
The climate of the basin is dominated by semi-arid climate (BSh) and tropical wet and dry
climate (Aw/As) climate, which is characterized by low rainfall, high temperature, and high
evapotranspiration. Three climate regions can be found in the basin (Ayalew et al., 2022). The
Abijata-Ziway subbasin which is characterized by low annual average rainfall ranging from
400 mm to 860 mm with one peak in August. The second basin, Abaya-Chamo subbasin, is
receives higher annual precipitation ranging from 704 mm to 1200 mm with a bimodal climate
pattern with peaks in April and August. The third subbasin, Segen subbasin, is characterized
by a bimodal rainfall pattern with peaks in April and October and receives an average annual
precipitation ranging from 500 to 1100 mm. Temperatures within the basin exhibit a wide
range, from 10°C to 36°C, with the warmest temperatures measured on the Rift Valley floor
and frost-prone conditions in the Afro-alpine zone. The southern region of the Rift Valley is
both lower in elevation and warmer and drier than the other areas of the basin.



Figure 1| Topography of the Rift Valley Lakes basin as well as the river network, river and climate stations used for modelling.

The basin is also characterized by a complex and rugged topography with active faults, volcanoes and hot springs (Stamps et al., 2008). Based on the digital elevation model (USGS,





2023) data, the slope ranges from 0 to 161 %. Agricultural land is the most dominant land use
followed by semi-natural vegetation and water bodies. Two lakes were chosen from the basin
as representative examples of its hydrological system, based on their degree of impact. Lake
Abijata was selected as an example of a monomodal climate regime, while Chamo Lake was
chosen to represent a bimodal climate regime. Lake Abijata is a terminal lake of the Ziway-
Abijata subbasin and linked to Lake Ziway via the Bulbula River and to Lake Lugano via the
Hurakolo River. Lake Chamo is a terminal lake for the Abaya-Chamo subbasin and linked with
Lake Abaya via the Kulfo River.
2.2.   Spatial and hydroclimatic data
The main hydrological characteristics of the basin have been acquired from the climate data,
topography, land use, soil properties, streamflow, lake level and bathymetry survey. Time
series of climate data for 44 stations (1981- 2018), a georeferenced Landsat image (1999),
bathymetry of Lake Chamo (1998 and 2021), streamflow of rivers (1987 – 2006), Lake level (
1987-2015) the Shuttle Radar Topography Mission digital elevation model (SRTM DEM)
(30x30) and soil data were used for this study. The SRTM DEM has been used, from which
slope, river network, and watershed boundaries are obtained. Selected soil physical properties
and the area coverage of each soil type were classified based on the requirements of the
hydrologic model. The bathymetry survey data was used to calculate the change in volume,
area and depth of a lake. It was also used to obtain information about sediment deposition in
the lake. Column
Table 1| List of data and source used in this study

| Data | Resolution(m) | Access Date | Sources |
|------|------------|-------------|---------|
| Landsat 5, Thematic Mapper | 30 | Feb 25,1999 | http://earthexplorer.usgs.gov |
| STRM DEM | 30 | 2014-09-23 | |
| Soil | 30 | 2016 | Ministry of water irrigation and electric city |
| Stream flow | - | 1987-2006 | |
| Lake level | - | 1987-2015 | |



| Climate ( rainfall and temperature) | - | 1987-2018 | Ethiopian National Meteorological Agency |
| Lake extent shape file | | 1999 | http://earthexplorer.usgs.gov |
| Bathymetry survey | - | 1998 & 2021 | AMU-VLIR-IUC |

## 2.3. Methodology

**Hydrological regime change analysis**

In this study, a common approach for analyzing hydrological regime change is utilized by combining a change point analysis and Indicators of Hydrological Alteration (IHA). This allows for a more robust and comprehensive analysis of hydrological regime change (Zhang et al., 2016, Vieceli et al., 2015, Vu et al., 2019). Change point analysis is a statistical method that is used to detect changes in the mean or variance of a time series. We used it to identify breakpoints of the Lake level. Once the break points have been identified, IHA were used to evaluate the degree of alteration of the natural flow regime. The Range of Variability Approach (RVA), Environmental Flow Components (EFC), Flow Duration Curve (FDC), Box-and-Whisker (BAW), and Percentile analysis of the IHA were used to evaluate the alteration of the hydrological regime (Gunawardana et al., 2021, Song et al., 2020).

The Range of Variability Approach (RVA) evaluates hydrological alterations by analyzing 33 hydrological parameters, which are categorized into five groups which evaluate the magnitude, timing, frequency, duration, and rate of change (Richter et al., 1998, Shieh et al., 2007) (Table 2). Group 1: The 12-monthly mean flows were computed to determine the water level. Group 2: twelve parameters were used to explain the range of changes in annual extreme flows in magnitude and duration in daily, weekly, monthly, and seasonal cycles. Group 3: Julian dates for 1-day annual maximum and minimum were computed to determine the variation of timing of annual extreme flows that can be associated with extreme conditions, such as floods or droughts. Group 4: four parameters are categorized in this group that refers to the frequency and duration of high and low pulses. The high pulses refer to periods within a year when the daily flows are above the 75th percentile of the pre-impact period. The low pulses are periods within a year when the daily flows are below the 25th percentile of the pre-impacted period. Group 5: three parameters are also categorized in this group to understand the direction and





magnitude of hydrological regime changes, including both positive and negative changes.
These parameters are important because they capture the variability of hydrological flow over
short periods, as high flow can cause erosion and sediment transport, while low flow pulses
can lead to habitat degradation and changes in water quality.
Table 2| Hydrologic parameters used in the RVA, and their features

| General group | Regime features | Streamflow parameters used in the RVA |
|---|---|---|
| Group 1: Median of monthly water condition indices | Magnitude, timing | The median monthly value for each month |
| Group 2: Yearly extreme water conditions indices | Magnitude, duration | 1-day minimum<br>3-day minimum<br>7-day minimum<br>30-day minimum<br>90-day minimum<br>1-day maximum<br>3-day maximum<br>7-day maximum<br>30-day maximum<br>90-day maximum<br>Number of zero days<br>Base flow index |
| Group 3: Yearly extreme water conditions indices | timing | Date of minimum<br>Date of maximum |
| Group 4: High and Low pulses indices | frequency and duration | Low pulse count<br>Low pulse duration<br>High pulse count<br>High pulse duration |
| Group 5: Water condition changes indices | rate and frequency | Rise rate<br>Fall rate<br>Number of reversals |

In the annual average water balance, the flow signals are averaged and it is difficult to
distinguish which flow signals affect the annual water balance. To avoid this limitation, we
used FDC to evaluate the variation of streamflow for the pre-impacted and impacted periods.
The shape of the curve is an index of the natural storage in the watershed, including the
groundwater. Since the dry season flow consists entirely of return flow from the groundwater,
i.e., the lower end of the FDC indicates the general characteristics of shallow aquifers.
Therefore, flow signals were determined from the FDC based on the exceedance probability
threshold. Here, low flow signals are defined as $\geq$ 75% of the exceedance probability, while
higher flow signals are defined as $\leq$ 20% of the exceedance probability and the rest are mid





flows. The choices of thresholds for disaggregation were based on earlier studies (Pfannerstill
et al., 2014, Smakhtin, 2001).
2.3.1. Hydrologic model
The Soil and Water Assessment Tool Plus (SWAT+), a completely revised version of the
SWAT model (Arnold et al., 1998) was built to simulate water balance fluxes in a watershed.
It is more flexible than SWAT in terms of the spatial representation of interactions and
processes within a watershed (Bieger et al., 2017). It simulates the hydrological processes from
precipitation to streamflow using the water balance equation. The variable storage routing
method was used for river routing and SCS curve number infiltration method was used. The
Hargraves equation was used to calculate the potential evapotranspiration, which is commonly
used in data scarce regions (de Sousa Lima et al., 2013, Moeletsi et al., 2013).
Currently, SWAT+ is tested in few watersheds across the world, where the results from
SWAT+ are favorable compared to the previous model version (Wagner et al., 2022)  It can
simulate the quantity and quality of water resources from a hydrological response unit to basin
scale. SWAT is also suitable to assess the impact of climate change (Mahmoodi et al., 2021a),
land cover change (Tigabu et al., 2019) and  land use change (McGinn et al., 2021, Wagner et
al., 2023), and watershed management practices on water resources (Mahmoodi et al., 2021b)
Moreover, it has been proven capable of modeling in data scarce regions (Tigabu et al., 2023,
Wagner et al., 2012). To depict the spatial heterogeneity of a watershed, each watershed is
divided into multiple homogenous hydrological response units based on a unique combination
of land-use, slope and soil characteristics (HRU).
2.2.1. Model parameterization, calibration, and validation
Sensitive parameters were adopted from previous research (Ayalew et al., 2023) for streamflow
simulation in two subbasins (Table 3). Lakes, which are located on the main channel network
of the subbasin, are accounted for during modelling. The lakes areal extent are used as input
for the SWAT+ model. The default model parameters for the reservoirs are used, e.g. SWAT+
assumes an average depth of 10 m to calculate the volume of the lake. During the calibration
process, the lake depth was modified using bathymetric data obtained from Ethiopian ministry
of water and energy.



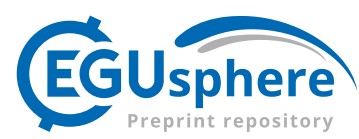

1  *Table 3* | Calibration parameters and the upper and lower boundaries used for calibration

| Parameters | Description | Limit | | Change | Fitted range & value | | | | | |
| --- | --- | --- | --- | --- | --- | --- | --- | --- | --- | --- |
| | | | | | Lake Abijata | | | Lake Chamo | | |
| | | Min | Max | | Min | Max | Value | Min | Max | Value |
| CN2 | Condition II curve number | -15 | +15 | abschg [a] | -15 | -5 | -1.8 | -15 | -10 | -11.8 |
| Sol-Awc | Available water capacity of the soil layer (mm H2O/mm soil) | -0.25 | +0.25 | abschg | -0.16 | +0.15 | -0.05 | -0.16 | +0.15 | -0.12 |
| ESCO | Soil evaporation compensation coefficient | 0 | 1 | absval [b] | 0.01 | 0.5 | 0.34 | 0.01 | 0.5 | 0.21 |
| SURLAG | Surface runoff lag Coefficient(days) | 0 | 24 | absval | 0.1 | 10 | 6.7 | 0.1 | 10 | 8.4 |
| PERCO | Percolation coefficient (mm H2O) | 0 | 1 | absval | 0.01 | 0.95 | 0.91 | 0.01 | 0.3 | 0.17 |
| LATQ_CO | Lateral flow contribution to reach (mm H2O) | 0 | 1 | absval | 0.01 | 0.95 | 0.63 | 0.01 | 0.3 | 0.10 |
| ALPHA_BF | Baseflow recession constant fast aquifer (days) | 0 | 1 | absval | 0.01 | 0.6 | 0.27 | 0.01 | 0.3 | 0.07 |
| k | Saturated hydraulic conductivity (mm h$^{-1}$) | -45 | +45 | pctchg [c] | -10 | +15 | 1.86 | -10 | +15 | -3.7 |
| EPCO | Plant uptake compensation factor | 0 | 1 | absval | 0.6 | 0.9 | 0.8 | 0.6 | 0.9 | 0.85 |
| z | Soil depth (mm) | -45 | +45 | pctchg | -15 | +0 | -3.7 | -15 | +0 | -7.5 |
| Lake depth | Average depth of a Lake (m) | 10 (default) | | absval | 2.5 | 9 | 5.5 | 5 | 13 | 9.7 |

2  Where, a abschg [a] adds an absolute value to the initial parameter value; b absval[b] replaces the initial parameter value with an absolute value;

3  c pctchg[c] increases or decreases of the initial parameter value by the given percentage of the value.





The streamflow data at each outlet (at each gauge from 1987 to 2006 splatted in to three periods
for model warm-up (one year), model calibration (1987 to 1995) and validation (1995 to 2006).
For calibration, 5000 parameter sets were generated with Latin Hypercube Sampling (Soetaert
and Petzoldt, 2010) using the parameter ranges given in Table 3. The model configurations
were evaluated for the same 5000-parameter sets. For each parameter set a model run was
performed and the final parameter fitted range was selected based on the best combined
$0.6 \leq KGE \leq 1$ (Gupta et al., 2009), $0.5 \leq NSE \leq 1$(Nash and Sutcliff, 1970), and percent Bias
$-25 \leq (PBIAS) \leq 25$, and $0 \leq$ RMSE-observations standard deviation ratio (RSR)$\leq 1$
values, so that the KGE , NSE , PBIAS and RSR were filtered within this range and the model
runs were accepted within these range. The best optimal value (fitted value) has selected among
the best value of KGE.  During calibration and validation, the water balance (total amount,
distribution through time), storm sequence (time lag or shift), and shape of hydrograph (rising
limp, peak, recession) are considered as key components. Calibration and validation were
carried out in R using the packages FME for Latin Hypercube Sampling (Soetaert and Petzoldt,
2010), hydroGOF for model evaluation (Zambrano-Bigiarini, 2018), and the packages zoo
(Zeileis and Grothendieck, 2005) and xts (Ryan and Ulrich, 2011)  for data processing.
In addition to statistical performance evaluation, visual inspection of the hydrography and flow
duration curve (FDC) provide information about the overall qualitative match between
measured and modeled streamflow.

## 20   2.2.2. Regionalization

A multisite (stepwise and pooled) regionalization method was applied to ensure that the model
accurately represents the hydrological behavior of multiple locations and that the model can be
used to make predictions for other locations. It improves the accuracy of the model by
calibrating it with data from multiple sites (Hundecha et al., 2016). The method can also help
to reduce uncertainty and improve the robustness of the model by taking into account the
variability of hydrological conditions across different sites.
Hydrological processes are primarily controlled by rainfall, topography, soil characteristics,
and land use, similarity between watersheds has been demonstrated prior to regionalization.
The stepwise calibration carried out on a nested class of sub-basins for the corresponding
multiple gauging stations in the sub-basin (Tsegaw et al., 2019). Multi-gauge calibration
strategy involves calibrating all parameters of the model domain simultaneously against
multiple streamflow gauges within the subbasin (Wi et al., 2014). This approach aims to look





for suitable parameters that are able to produce satisfactory model results at all neighboring
gauging stations in a single implementation of optimization. Therefore, pooled regionalization
technique has been chosen for this study based on its ability to predict streamflow and
calibration uncertainties. To do so, Bulbula assumed to be gauged watershed and Katar at
Abura and Meki were assumed ungauged watersheds for validation (Figure 2). Likewise, Kulfo
at Abaya has considered as gauged watershed and Bilate, Gidabo and Upper Gelana considered
as ungauged watershed (Figure 2). The gauged stations are considered as the "donor"
watershed, and the ungauged stations are considered as the 'pseudo' watersheds. The donor
watersheds are used to parameterize and calibrate the SWAT+ model parameters. The pseudo-
ungauged watersheds are used to test the calibrated model. In testing the model on the pseudo
ungauged watersheds, the model was run with calibrated parameters for the same period and
the model performance for donor and pseudo watersheds were evaluated using the objective
functions KGE, NSE, PBIAS and RSR. If the developed model for the donor watershed
performed sufficiently (KGE$>$ =0.5, NSE=0.5, -25 $<=$ PBIAS $<=$ and 0 $<=$ RSR $<=$ 1) for the
pseudo watershed, the model is considered robust and was used for regionalization. The
optimal values of parameters of each catchment were transferred to the nearest ungauged
catchments and reach.



2      Figure 2| Multisite-pooled calibration technique to estimate flow of ungauged watersheds



2.2.3.  Bathymetry analysis
The Lake Chamo was surveyed using Multibeam echosounder (MBES). It is sonar technology
that uses a band of multiple beams to create a high-resolution 3D map of the lake bed. It is
based on the principle of emitting a series of acoustic pulses and measuring the time it takes
for the sound to travel to the lake bed and back. Although our hydrologic model was developed
on two terminal lakes, Abijata and Chamo, we conducted a bathymetry survey only for Chamo
Lake.
The survey was conducted during the dry season from 1 to 21 July 2021. Before the survey
began, the surface elevation of the lake level was measured with a handheld GPS, which was
at a level of 1110 m and used as the reference level. Following that, a continuous record of lake
floor topography was measured with an MBES mounted on a motor boat at a constant speed
of 5 km/h along predefined average traverse lines at 450 m spacing along a north-south
direction.  A crossing and recrossing survey was also performed in the same area to correct
errors caused by potentially misleading reflections and variations in the speed of sound moving
through the lake (Figure 3). To supplement the dataset produced by the echo sounder, zero
depth coordinates were taken simultaneously with hand-held GPS along the lake's shoreline at
shorter distances. Because there were no other obstacles, the true size of the lake was preserved
during mapping using these border coordinates. The bathymetry data typically in the form of
point data (XYZ) that represent the depth measurements at various locations within the lake
was imported into Surfer20 to analyze the lake floor morphology in 2D and 3D, as well as
depth, area, and volume.



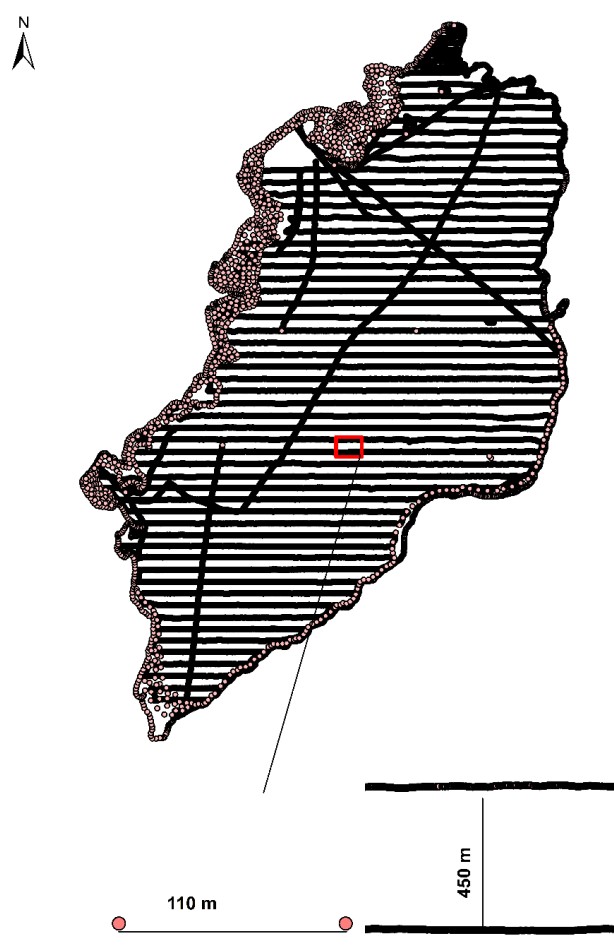

2  Figure 3| Bathymetry survey using multibeam echosounder (MBES) along predefined average
3  traverse lines at 450 m spacing along a north-south direction and 110 m along east-west
4  direction.



Results and Discussions
3.1.    Lake water balance analysis
Hydrological modelling and model performance
The hydrographs shown in Fig. 4 indicate that the SWAT+ model was successfully calibrated
and validated for each 'donor watershed'. Similarly, the hydrograph depicted in Fig. 5 signifies
the model's success within each respective 'pseudo' watershed. The relationship between the
simulated and observed streamflow based on the model is statistically summarized in Table 4.
The values of the objective functions indicate a strong goodness-of-fit for both 'donor and
pseudo watersheds.

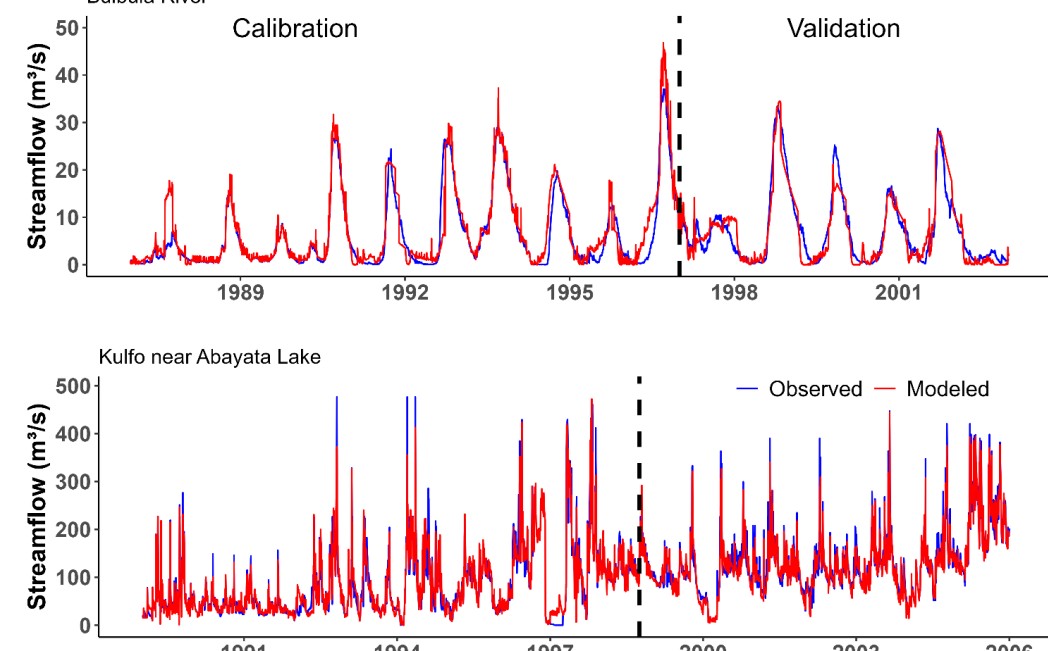

Figure 4 | Hydrograph of modeled and observed daily streamflow for calibration and validation
period of the donor watersheds.





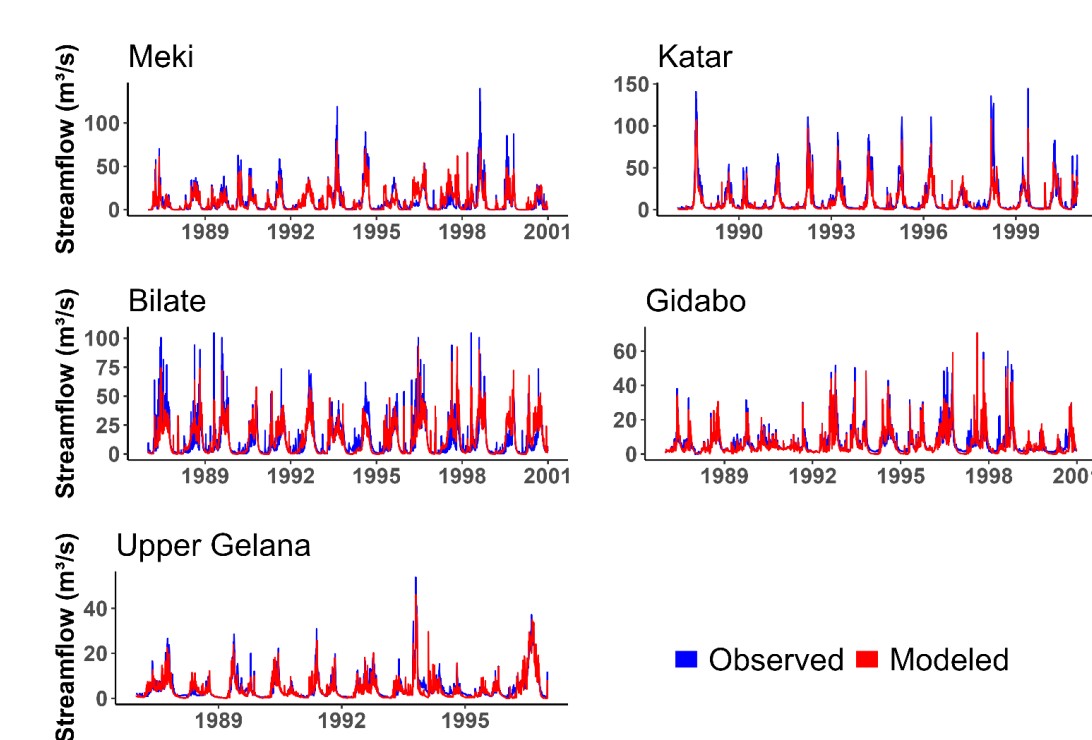

Figure 5 | Hydrograph of modeled and observed daily streamflow of the 'pseudo' watersheds
Similarly, the FDC in Fig. 6 & 7 show that there is a good agreement between observed and
the modeled discharge in both 'donor and pseudo' watersheds. Throughout the calibration and
validation period, the model shows a slight overestimation of high, middle and low flows in
Bulbul River. However, at Kulfo, high and low flows are slightly underestimated. The test of
the regionalization approach shows that high flows were underestimated in all 'pseudo'
watersheds.





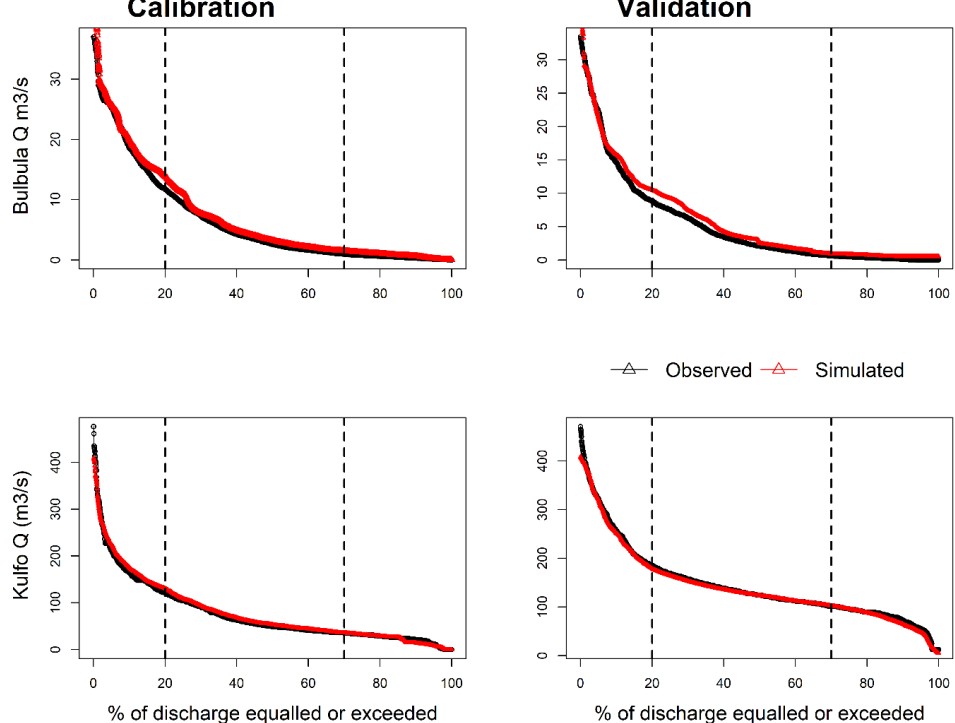

2 Figure 6 | Flow duration curves of observed and modeled daily streamflow for the calibration
3 and validation period of the 'donor watersheds.





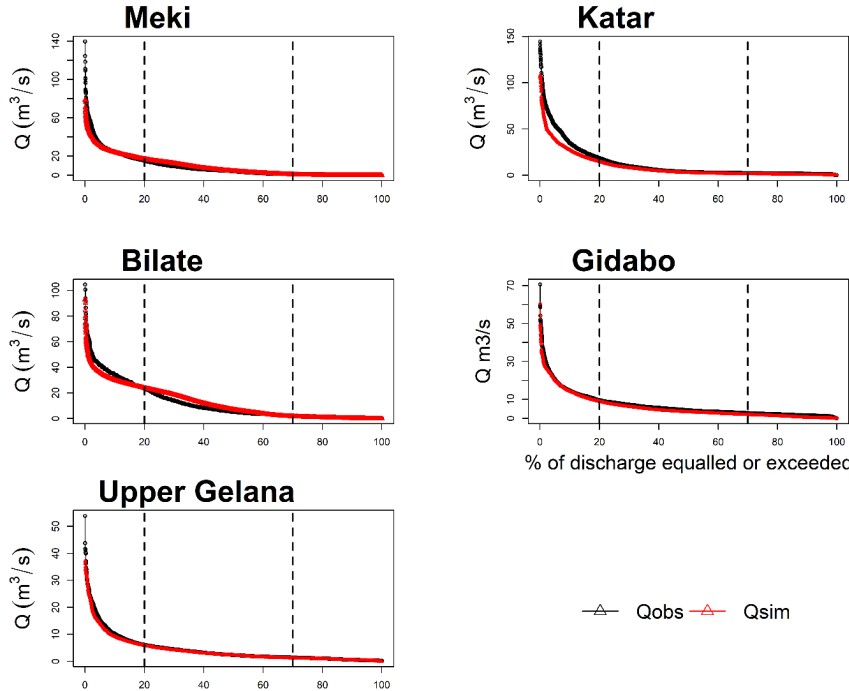

Figure 7 | Flow duration curves of observed and modeled daily streamflow of the 'pseudo
watersheds.
Table 4 | Model evaluation statistics of the daily SWAT+ for the calibration, validation and
regionalization (1987-2000)

| Watershed | Method | Objective functions | | | |
|---|---|---|---|---|---|
| | | KGE | NSE | PBIAS | **RSR** |
| **Bulbula (Donor)** | Cal. | 0.84 | 0.76 | -2.1 | **0.51** |
| | Val. | 0.80 | 0.72 | -5.4 | **0.53** |
| **Meki (Pseudo)** | Test | 0.71 | 0.63 | 9.2 | **0.69** |
| **Katar (Pseudo)** | Test | 0.75 | 0.64 | 10.7 | **0.67** |
| **Kulfo@ Abayata (Donor)** | Cal. | 0.81 | 0.68 | 3.7 | **0.56** |
| | Val. | 0.78 | 0.63 | 7.6 | **0.68** |
| **Gidabo (Pseudo)** | Test | 0.70 | 0.59 | +9.3 | **0.75** |
| **Upper Gelana (Pseudo)** | Test | 0.67 | 0.61 | +11.8 | **0.73** |
| **Bilate (Pseudo)** | **Test** | **0.59** | **0.53** | **+14.3** | **0.78** |



Lake water balance change
The water balance of Lake Abijata is controlled by rainfall and evaporation, water abstraction
from the lake by the soda ash factory, inflow from Bulbula and Horakelo Rivers, and inflow
from ungauged watersheds. The water balance of Lake Chamo is primarily controlled by
evaporation and rainfall on the lake, inflow from Kulfo River, 40 springs, and inflow from
ungauged watersheds (Sile and Elgo) (Table 5). We assumed that the groundwater flow and
movement influence is negligible, following findings by Ayenew (2004). The result depicted
that the surface runoff and evaporation increased in the impact period in both lakes. Abstraction
increased in the Abijata Lake (Table 5). Changes in lake level and volume reflect variations in
the inputs from rainfall, evaporation, abstraction and surface flow. Trends in the lake water
levels in both lakes are highly variable. The Abijata Lake level decreased by 2 m due to large-
scale water use for irrigation and soda abstraction in the catchment, significant changes have
been recorded in the past few decades, and by 2021, the area of Lake Abijata had been reduced
by about 68% as compared to year 1989( Ayalew et. al, 2022) and 60%  as compared to the
year 2016 (Wagaw et al., 2019). Because irrigation is highly expanding and abstraction of water
from Meki, Katar and Bulbula Rivers, and Lake Ziway for irrigation takes place year-round,
its effect on water levels is increased during dry season (Getnet et al., 2014, Jansen et al., 2007,
Wagaw et al., 2019). The water balance of the lakes in Table 5 was determined by combining
model output, including lake areal rainfall, evaporation, and inflow from both gauged and
ungauged watersheds, with recorded data such as abstraction, depth, and spring inflow.
Table 5| Changes of the annual water balance components in million m³

| Water Balance Components | Lake Abijata | | | Lake Chamo | | |
|---|---|---|---|---|---|---|
| | Pre-impact period | Impact period | ΔV, million m³/y | Pre-impact period | Impact period | ΔV, million m³/y |
| Lake areal rainfall | 103.3 | 96.6 | -6.7 | 338.4 | 329.4 | -9.0 |
| Lake evaporation | 240.7 | 233.9 | -6.9 | 496.2 | 518.3 | 2.2 |
| Gauged river inflow | 238.3 | 208.4 | -29.8 | 2184.3 | 3943.0 | 1758.7 |
| Inflow from springs | - | - | - | 5.3 | 4.1 | -1.2 |
| Ungauged river inflow | 14.7 | 16.3 | 2.4 | 171.6 | 312.1 | 140.5 |
| Lake outflow | - | - | - | - | - | - |
| Abstraction | 5 | 13 | 8 | - | - | - |
| Enclosure term | 110.6 | 74.4 | -35.2 | 2203.4 | 4070.3 | 1886.8 |
| Change in depth (m) | 4.3 | 2.7 | -1.6 | 14.2 | 18.6 | 4.4 |



## 3.2. Hydrological alteration

The analysis of hydrological alterations was carried out on both Lake Abijata and Lake Abaya. The change point analysis presented in Fig. 6 shows that there is an abrupt shift in lake level over the Rift Valley Lake Basin. The change point analysis results revealed that the lake level has decreased after the year 2003. Abaya Lake has experienced a decrease after a change point from the 2000s until 2009 and more recently, a large increase was observed. Based on the change point analysis, three periods have identified, the pre-impact period (1985-2003), impact period (2003 to 2009) and post-impact period (2010 to 2015).

The environmental flow analysis in Fig. 6 depicted that the natural flow pattern has been disrupted. The magnitude and timing, duration, seasonality and frequency of high and low flows of the impacted periods have changed compared to the pre-impact period. The extreme low flow, low flow, small floods and large flood values for the impacted period have decreased in Abijata Lake, while they have increased in the Abaya Lake.





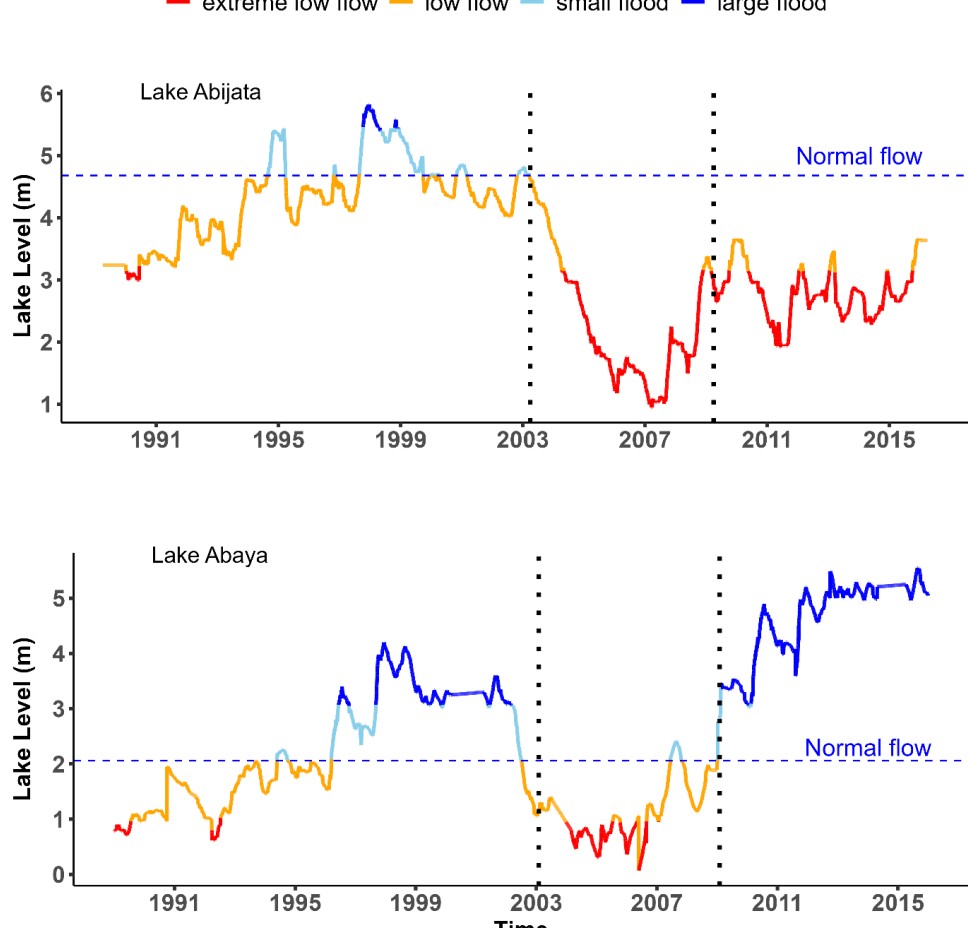

Figure 8 | Environmental flow analysis of Lake Abijata and Abaya based on observed data
Fig. 9 depicts the monthly median and FDC plot of Lake Abijata and Abaya in the pre-impacted
and impacted period. It is observed that the lake level and the natural regime have altered after
the year 2002/2003 in both lakes. The Abijata lake level exhibited a significant decrease during
the pre-impacted period for all months, while the Abaya lake level showed a significant
increase during the impacted period in all months. The FDC reveals that, again, the Abaya lake
level has increased in all flow regimes (low, middle and high), while has decreased in Abijata
lake. A high degree of alteration is observed in both lakes; but the hydrological regime of Lake
Abaya has undergone a complete change, especially during the dry season (May to September).
This is associated with a change of the rainfall pattern (Ayalew et al., 2022).



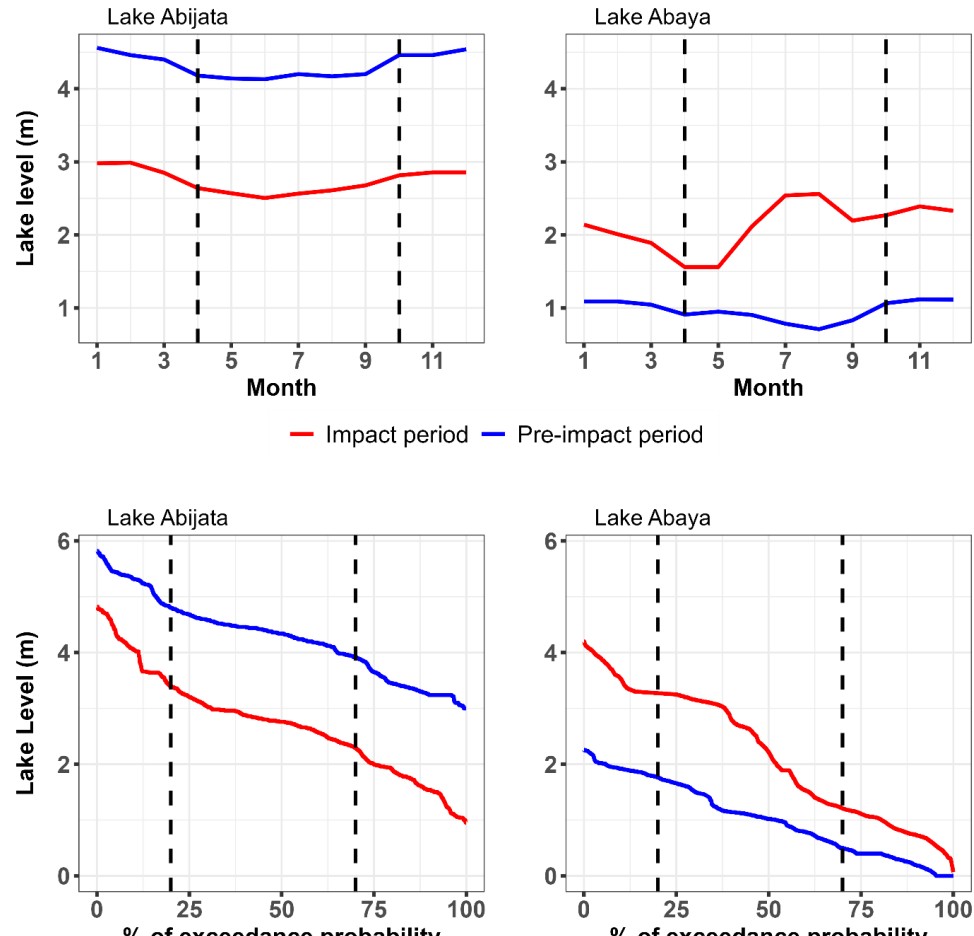

Figure 9 | Monthly median and FDC analysis of pre-impacted and impacted period

Median of monthly flow (RVA Group 1) throughout the impact period indicates a decreasing trend compared with the pre-impact period in the Abijata Lake. The dispersion coefficients for the pre-impact period (ranging from 0.21 to 0.33) are mostly lower than those for the impact period (ranging from 0.3 to 0.49), indicating the higher flow fluctuations in the impacted period due to the irrigation expansion and soda ash factory. Whereas in Abaya Lake, median of monthly flow depicted an increasing trend compared with the pre-impact period. The dispersion coefficients for the pre-impact period (ranging from 1.18 to 1.68) are mostly lower than those for the impact period (ranging from 0.80 to 1.5), indicating that the higher flow fluctuations in the impacted period due high runoff are associated with high deforestation. The



medians of annual 1-, 3-, 7-, 30-, 90-day minimum and 1-, 3-, 7-, 30- and 90-day maximum for
the impact period decrease significantly for Abijata Lake and significantly increase in the case
of Abaya Lake. These results indicate that the daily, weekly, monthly and seasonal maxi-
mum/minimum flow cycles are negatively influenced by water abstraction for irrigation and
factory in the case of Abijata Lake; and positively influenced by runoff increasing the case of
Lake Abaya.
The median Julian dates for the annual 1-day minimum have shifted forward in the impacted
period, moving from the 137th and 91st day in the pre-impact period to the 130th and 113th
day in Lake Abijata and Abaya, respectively. Likewise, the median Julian dates for the annual
1-day maximum have also moved forward in the impact period, shifting from the 309th and
320th day to the 327th and 311st day in Lake Abijata and Abaya, respectively. The result also
showed that there is no significant change on the medians of low, high pulse counts and base
flow index in the impact period and in the pre-impact period, in both lakes (Figure 10).
The medians of low pulse durations (increased from 2 to 47) and high pulse durations
(increased from 64 to 140) in the impacted period are   higher than those in the pre-impact
period in Abijata Lake. However, the medians of low pulse durations (decreased from 125 to
36) and high pulse durations (increased from 208 to 223) in the impact period in Abaya Lake
which indicates a high hydrologic alteration of low and high pulse durations.  Median of high
pulse durations have increased in both lakes, which is associated with increasing runoff during
wet season.
The medians  of rise rate and  fall  rate are the same as those in the pre-impact period in
both Lakes except the number of reversals. The number of reversals are higher than in the pre-
impacted period (decreased from 41 to 23) in Abaya lake; and (decreased from 88 to 78) in
Abijata lake.
Considering all 33 parameters (Figure 10), the highest hydrologic alteration factors has
occurred in low RVA category (ranging from 1 to 2.7) except fail rate and high pulse duration,
which indicates annual parameter values more often fell inside the RVA target window than
expected in Abijata Lake. Likewise, in this lake the second highest hydrologic alteration factors
have occurred in high RVA category (ranging from -1 to -0.1) except low pulse count, which
indicates annual parameter values less often fell inside the RVA target window than expected.
High hydrologic alteration factor has also been observed in Abaya Lake which in the low RVA
category (ranging from -1 to -0.1) except fail rate and low pulse duration, which indicates





annual parameter values less often fell inside the RVA target window than expected. In this
lake also the second highest hydrologic alteration factor has occurred in high RVA category in
group one and group two (ranging from 0.6 to 1.5), which indicates annual parameter values
more often fell inside the RVA target window than expected.



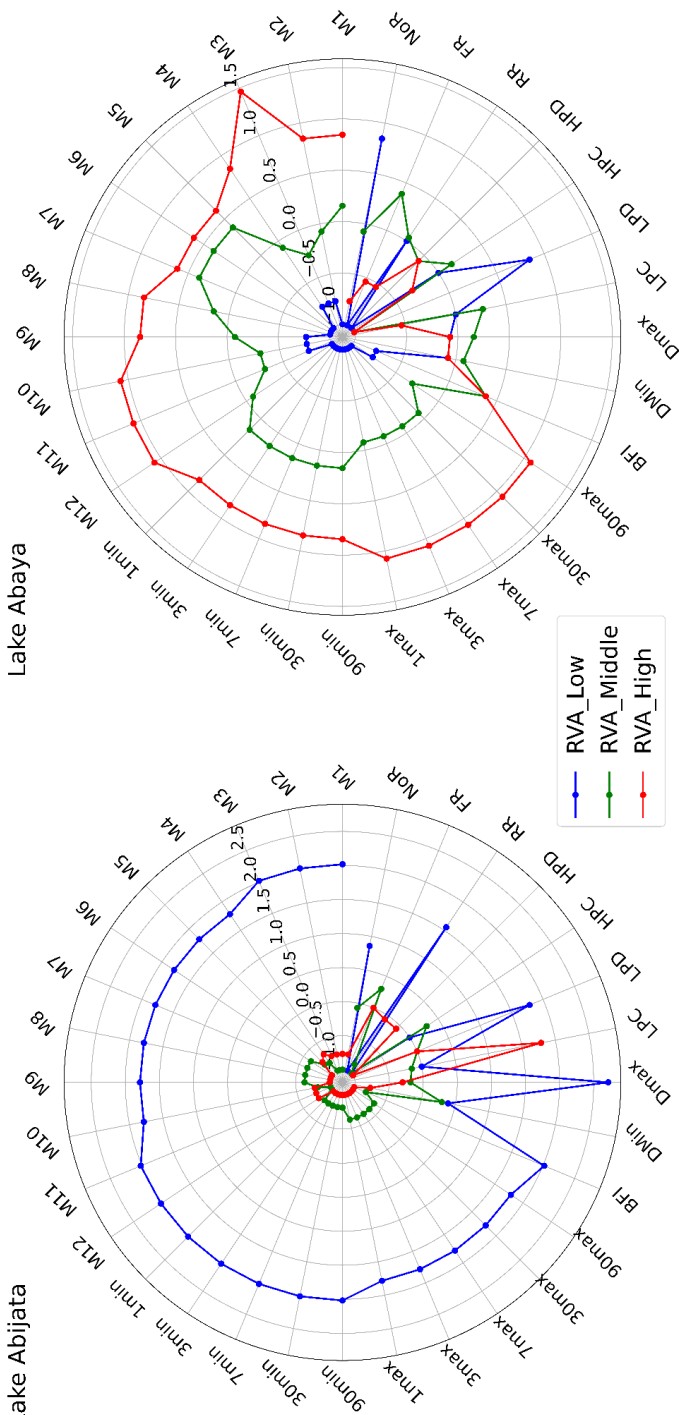

Figure 10| Hydrological alteration analysis using RVA

Where, 'M1', 'M2', 'M3', 'M4', 'M5', 'M6', 'M7', 'M8', 'M9', 'M10', 'M11', 'M12', '1min', '3min', '7min', '30min', '90min', '1max', '3max', '7max', '30max', '90max', 'BFI', 'DMin', 'Dmax', 'LPC', 'LPD', 'HPC', 'HPD', 'RR', 'FR', 'NoR' are January', 'February', 'March', 'April', 'May', 'June', 'July', 'August', 'September', 'October', 'November', 'December', '1_day_min', '3_day_min', '7_day_min', '30_day_min', '90_day_min', '1_day_max', '3_day_max', '7_day_max', '30_day_max', '90_day_max', 'Base_flow_index', 'Date_of_min', 'Date_of_max', 'Low_pulse_count', 'Low_pulse_duration', 'High_pulse_count', 'High_pulse_duration', 'Rise_rate', 'Fall_rate', 'Number_of_reversals'.






Generally, results indicate that the hydrological system is changing in both lake basins but
responding differently to the changes. Changes in hydrology are mainly associated with human
activity and climate change. Apparently, the changing hydrology for the impacted period
(2003-2009) was worsen in the Abijata Lake and linked with frequent droughts and water
abstraction. Despite the fact that the Abijata Lake's level increased during the post-impacted
period, it remained lower than the lake level observed in the pre-impacted period. In contrast
to the pre-impacted period, the water level of Lake Abaya increased during the post-impacted
period.
Bathymetry analysis, Bathymetric characteristics of Lakes
**Morphometric characteristics and contour maps**
Morphological parameters used to characterize the morphometry of the Lake Chamo, including
area (A), Volume(V), maximum effective length (Lme), maximum width (Wme), mean width
(W), maximum depth($d_{max}$) and mean depth ($d_{mean}$), are summarized in Table 6. The values are
presented with respect to the reference datum discussed in the methodology. The result is also
compared with the bathymetry analysis of Awulachew (2001) conducted in 1998. The
maximum depth increased by 4.4 m in the middle of the lake , the sediment thickness increased
by about 0.6 m,  the area increased by 30 km$^2$ and the volume increased by 7.8 x $10^8$ m$^3$ within
two decades (Figure 8).
Table 6 | Morphometric characteristics of Lake Chamo

| Parameter | Chamo Lake | |
|---|---|---|
| | This study (2021) | Awulachew (1998) |
| Altitude (m) | 1110 (GPS) | 1107(EMA 1: 50 000 maps) |
| Basin area, including lakes (km$^2$) | 18 599.8 (with Lake Abaya contribution) | 18 599.8 (with Lake Abaya contribution) |
| A, including islands (km$^2$) | 346.76 | 316.72 |
| Volume (m$^3$) | 4.12× $10^9$ | 3.24 × $10^9$ |
| L$_{me}$ (km) | 33.93 between 5°41′36″N and 37°28′40″E to 5°58′24″N and 37°35′56″E | 33.50 between 5°42′00″N and 37°39′00″E to 5°58′00″N and 37°36′00″E |
| W$_{me}$ (km) | 16.07 | 15.5, perpendicular to L |
| W (km) | 10.17 | 10.1 |
| d$_{max}$ (m) | 18.6 | 14.2, near the middle |
| dmean (m) | 11.38 | 10.23 |

The water depth distribution obtained using Kriging interpolation method in surfer20 is shown
in Figure 9. A grid size of 110 m x 450 m bathymetric data was used to interpolate the depth



1 distribution of the lake. The resulting depth estimates were plotted on a contour map to

2 visualize the underwater topography of the lake; and we observed that the central region of the

3 lake had the deepest parts.



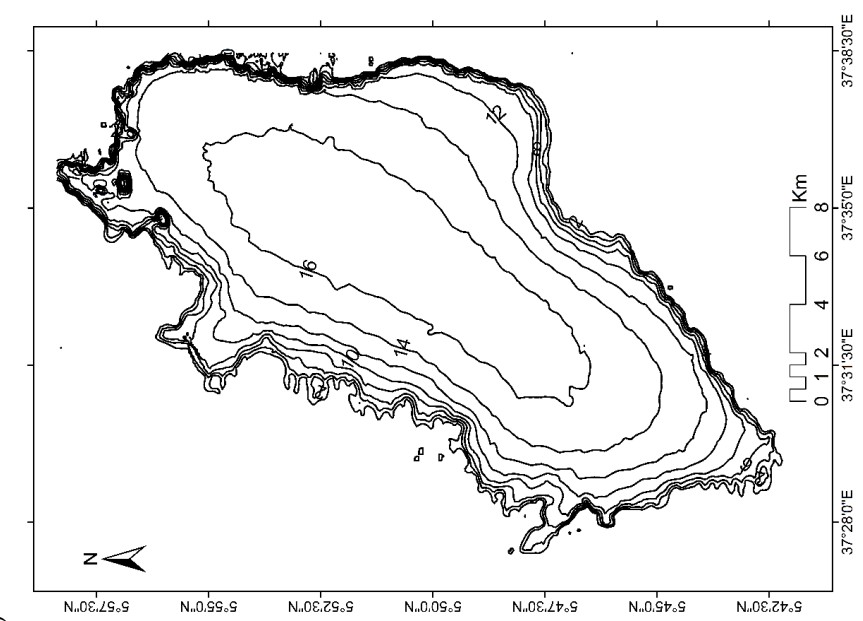

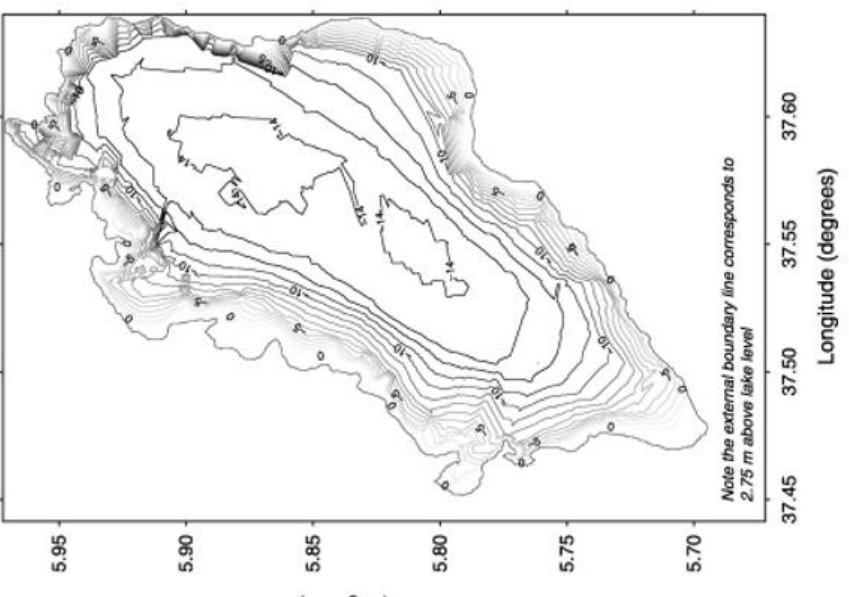

*Figure 11 | Bathymetric maps of Lake Chamo (a) Bathymetry of Awulachew (1998); (b) bathymetry of this study (2021)*





**Water level and storage change analysis**
Fig. 12 illustrates the changes in spatio-temporal patterns of Lake Chamo area and storage over
a span of 20 years (1998-2021), as indicated by the depth-area and storage analysis.
Specifically, the lake has experienced an expansion, gaining an area of 11.86 km$^2$, and its
volume has increased by 7.8 x 108 m$^3$ over the same period. As the maximum depth increased
by 4.4 m and the lakebed level increased by 0.6 m possible factors that could contribute to such
changes in a lake size and volume are increased runoff and sedimentation from rivers or streams
that flow into the lake due to land use change, deforestation. Another reason may be changes
in the underlying geology of the area, such as changes in the flow of underground water and
springs, that can contribute to changes in a lake's size and volume. These changes in size and
volume could have significant ecological and environmental impacts on the lake and its
surrounding area.

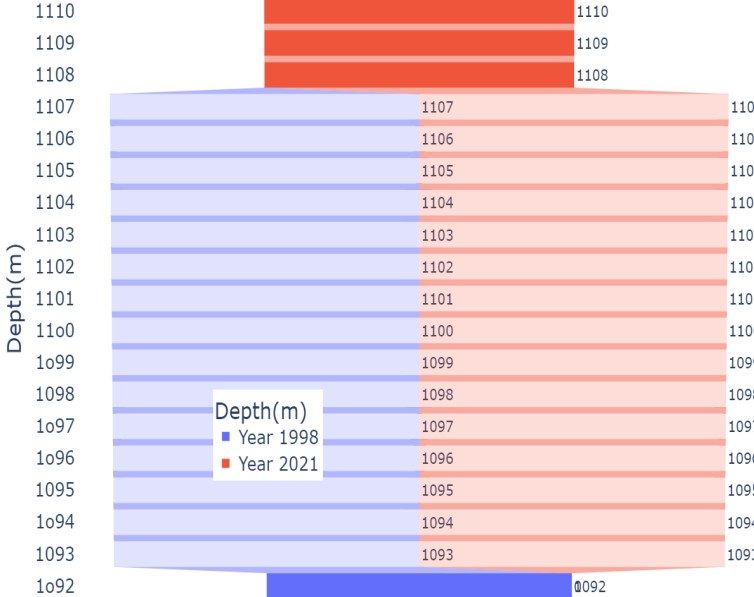





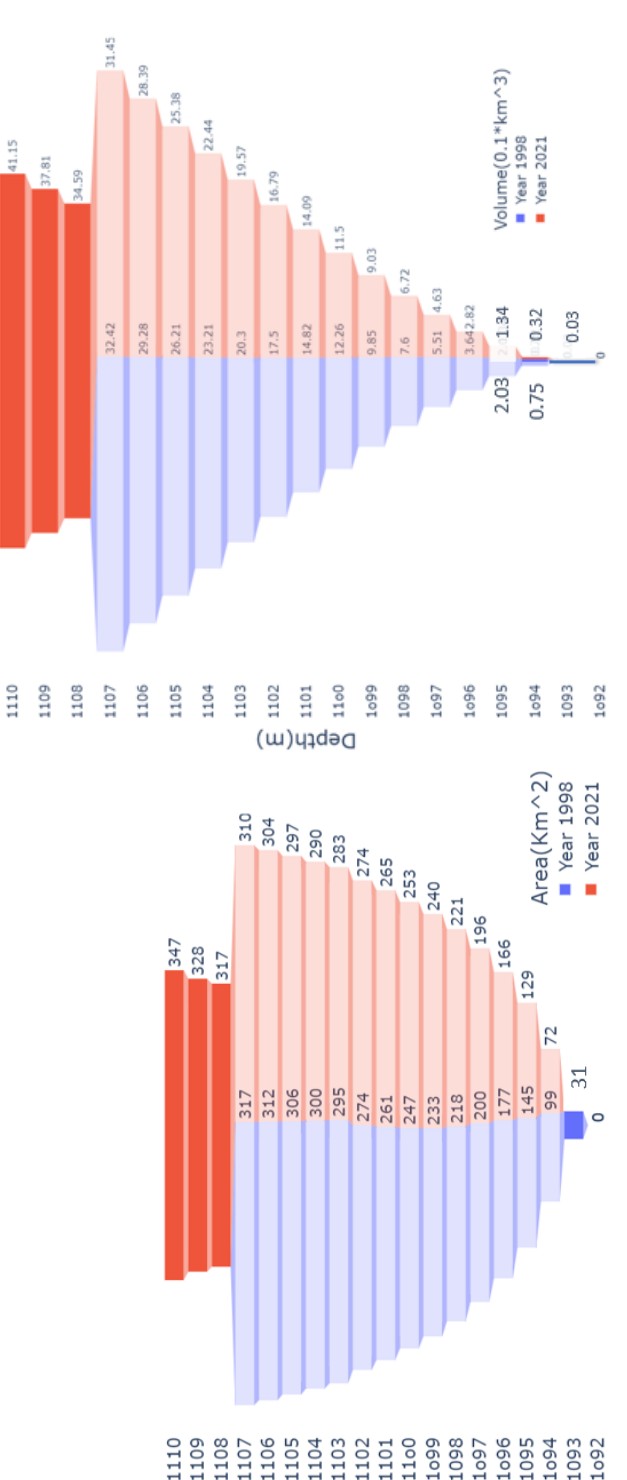

Figure 12 | Depth-Area-Volume change analysis: blue color for the 2021; red color for 1989 (Awulachew (1998)





Discussion
The hydrological system of the Rift Valley is highly complex and dynamic, as the region is
characterized by a series of interconnected lakes, rivers, wetlands, and groundwater systems
that collectively form a unique hydrological and ecological network. Additionally, the Rift
Valley is characterized by a complex topography, rainfall pattern and it is subject to geological
influences that contribute to its hydrological complexity.
The Rift Valley lakes possess distinct characteristics as they are structurally controlled and
form a unique hydrogeological system within the Rift. Their underground interconnection
through NE-SW aligned regional faults contributes to this uniqueness. Additionally, many of
these lakes, occupying volcano-tectonic depressions, are connected to one another through
local rivers. Due to the complexity of the system, relying on a single approach to understand
the hydrological system may not be effective. Therefore, employing a combination of
innovative methodologies helps to evaluate and understand the changing hydrological system.
The use of the hydrologic model SWAT+ allowed for a detailed analysis of lake dynamics,
while change point analysis and indicators of hydrological alteration (IHA) provided valuable
insights into the temporal changes in lake levels and inflows. Incorporating bathymetry survey
data enhanced the accuracy of the analysis by providing detailed information on lake depth and
sedimentation. Moreover, the regionalization (pooled calibration) approach extended the
applicability of the hydrologic model to ungauged watersheds, improving the generalization of
findings across the region.
In the context of regionalization of model parameters, our hydrological model exhibited a
performance level that aligns closely with the findings documented in the literature (Seibert,
1999, Beldring, 2002, Merz and Blöschl, 2004). Based on the evaluation using the objective
functions NSE, KGE, PBIAS and RSR, the pooled calibration technique showed a good
performance. It exhibited a remarkable ability to accurately represent observed hydrological
behavior, particularly for the closest pseudo watersheds (Gidabo, Upper Gelana, Meki, Katar,
and Bilate). This phenomenon of varying performance across spatially related watersheds has
also been noted by Merz and Blöschl (2004), who attributed it to the interplay of spatial
hydrologic variability and poor data quality. Seibert (1999) observed a decline in NSE runoff
efficiency from 0.81 to 0.79 when transitioning from calibrated to regionalized parameters
across 11 catchments and a decrease to 0.67 for a separate set of 7 catchments. Beldring (2002)
also found a NSE of 0.68 for 141 gauged catchments and 43 catchments treated as ungauged,





though approximately 20% of the latter group exhibited efficiencies below 0.3. Compared to
this the maximum decline in model performance due to regionalization in our study is observed
in the Bilate watershed with a decreade of the NSE from 0.68 (cal) /0.63 (val) in the donor
watershed to 0.53 in the pseudo watershed (Table 4). In this case the donor watershed (Chamo)
is dominated by semi-natural vegetation, whereas, the pseudo watershed (Bilate) has a larger
share of agricultural land. It is evident that while regionalization allows for broader model
applicability, the accuracy of predictions can vary significantly depending on contextual factors
and the specific characteristics of the studied watersheds.
The study revealed significant changes in lake levels and inflows over the past two decades in
the Rift Valley Lakes. Notably, Chamo Lake experienced substantial increases in area, depth,
and volume, while Lake Abijata witnessed an extraordinary decrease in area and depth. The
change point analysis revealed significant shifts in the hydrological system of Lake Abijata and
Lake Abaya after the year 2003. Substantial decreases in lake levels were observed in both
lakes from 2003 to 2009. The lake levels recovered after 2009. In the post-impact period (2010-
2015), the lake level of Abaya Lake showed an increase compared to both the pre-impact
(1987-2003) and impact period (2003-2009). On the contrary, during the same period, the lake
level of Abijata Lake showed a noticeable decreasing trend compared to the pre-impact period.
According to Ayalew et al. (2022), the significant decline in water level during the impact
period (2003-2009) was primarily attributed to prolonged drought affecting both lakes. A study
by Street (1979) also investigated that many of the rift lake's water level fluctuations were
associated with climate conditions rather than anthropogenic factors. However, recently the
changing water level is also associated with anthropogenic activities (Alemayehu et al., 2006).
The Range Variable Approach (RVA) analysis also revealed that the river's flow regime has
deviated significantly from its natural or historical patterns. The High-RVA and Middle-RVA
values are negative in Lake Abijata, which indicates that the river's flow conditions are lower
than what would be expected based on historical data. The negative value of high andmiddle
and the positive value of low level of Lake Abijata coincides with the time of high-water
abstraction for soda production and water abstraction for irrigation from the upstream Lake
Ziway. In the wet season, the time between May to September is a refilling period of the lake
from large inflows from the Katar and Meki rivers. During the dry season (November to June),
Ziway shows a net loss of storage due to high abstraction of water for irrigation. This high
abstraction of water leads to shifts in the flow regime, and a positive value of low-RVA as
observed. In Abaya Lake, the hydrological system responded reversely. The High-RVA and





Middle-RVA values are positive, and the low-RVA value is negative. This indicates that the
peak flow increased in the impact period during the wet season. Our previous work, Ayalew et
al. (2023), showed an increasing surface runoff and decreasing infiltration and evaporation
after 2000, which is associated with deforestation and agricultural expansion.
Lake Abijata is a terminal lake; it lacks any surface or groundwater outflow, making its water
level and volume subject to changes in hydrological budget components such as rainfall, river
inflow, and evaporation. However, recent development schemes in soda ash extraction and
irrigation have also contributed to the drastic reduction of the lake level (Alemayehu et al.,
2006). The main inflow is from the discharge from the Horakelo and Bulbula rivers, which are
the outflows of lakes Langano and Ziway, respectively, and direct rainfall. The total river
inflow decreased by 12.5%, which is associated with water abstraction for irrigation. As it is a
terminal lake, the ways of water loss from the lake are evaporation and abstraction. Therefore,
the main reason for hydrological regime change was water abstraction of feeding rivers for
irrigation and water abstraction for industrial purposes from the lake followed by evaporation.
Alemayehu et al. (2006) also state that abstraction of Lake Ziway for irrigation also has an
influence on the level of Lake Abijata. Conversely, Lake Chamo experienced an increase in its
water level during the post-impact period. The lake level increased by 4.4 m, and the lake
bottom elevation increased by 0.6 m. This significant water level rise is mainly due to high
surface runoff and sediment transportation. The total surface inflow increased by 80.5%, which
is influenced by changes in Land Use and Land Cover, particularly deforestation and
agricultural expansion, which resulted in higher runoff (Ayalew et al., 2023) and enhanced
sediment transportation. In contrast to many East African terminal lakes, lake Chamo has
shown significant water level rise.
Summary and conclusions
The Rift Valley Lakes in Ethiopia have experienced changes in their hydrological regime, due
to anthropogenic and climate change. Integrating a physical based hydrologic model, break
point analysis, indicators of hydrologic alteration (IHA) and bathymetry survey is a better
approach to understand a dynamic and complex hydrological system and the potential diving
factors. Indicators of hydrological alteration (IHA) were derived from lake level data that
clearly showed the alterations of hydrological regime. Main water balance components were
simulated using a semi-distributed Soil and Water Assessment Tool plus (SWAT+) model.
Multisite regionalization techniques represent the hydrological behavior of multiple locations





and that the model can be used to make predictions for other locations. The SWAT+ model
performed well for daily stream flow during calibration and validation period. The pooled
calibration approach showed a satisfactory performance in capturing observed hydrological
behavior, both in 'donor' and 'pseudo' watersheds. The applied calibration technique was
suitable for regionalization of model parameters as the rather small decrease of model
performance in the pseudo ungauged watersheds showed.
The findings reveal notable changes over the past two decades. Chamo Lake experienced an
increase in area by 11.86 km$^2$, depth by 4.4 meters, and volume by 7.8 x 10$^8$ cubic meters. In
contrast, Lake Abijata witnessed an extraordinary 68% decrease in area and a depth decrease
of -1.6 meters. Mean annual rainfall decreased by 6.5% in Abijata Lake and 2.7% in Chamo
Lake during the impacted period. Actual evapotranspiration decreased by 2.9% in Abijata Lake
but increased by up to 0.5% in Chamo Lake. Surface inflow to Abijata Lake decreased by
12.5%, while Lake Chamo experienced an 80.5% increase in surface inflow. Sediment depth
in Chamo Lake also increased by 0.6 meters.
The results of this study highlight on the changing hydrological regime in Chamo Lake,
emphasizing the role of anthropogenic influences in increasing surface runoff and sediment
intrusion. Conversely, the hydrological dynamics of Abijata Lake are primarily affected by
water abstraction from the rivers and lakes that serve as its water sources, driven by industrial
and irrigation demands. By examining these factors, this research offers valuable insights into
the evolving hydrological systems of the Rift Valley Lakes in Ethiopia, contributing to a better
understanding of the driving forces behind these changes.
The hydrological regime of Chamo Lake has experienced changes characterized by increased
area, depth, and volume, primarily influenced by heightened surface runoff and sediment
intrusion.
Abijata Lake has undergone significant changes in its hydrological regime, marked by a
substantial decrease in area and depth, resulting from water abstraction for industrial and
irrigation purposes from feeding rivers and lakes.
The findings emphasize the importance of understanding the driving factors behind
hydrological changes in Rift Valley Lakes, particularly the influence of anthropogenic
activities and climatic variations.



Further research and monitoring efforts are necessary to deepen our understanding of the
hydrological processes and identify effective management strategies to ensure the sustainable
use and conservation of the Rift Valley Lakes in Ethiopia.
**Acknowledgments:**
The German Academic Exchange Service (DAAD) and the Ethiopian Ministry of Education
supported this research. Our gratefully acknowledgment is also towards to Ethiopian Ministry
of water resources for providing hydrological data; Ethiopian Meteorology Agency for
providing meteorological data and Arba Minch University- Vlaamse Inter-universitaire Raad-
Institutional University Cooperation (AMU-VLIR-IUC) program for funding the bathymetry
survey conducted on Lake Chamo.

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
