# Peer review of "from the Ethiopian Rift Valley Lakes Basin"

_EGUsphere, 2023_

## Author Comment (AC1)

**Response to Anonymous Referees**

**Dear Editor Marleen de Ruiter ,**

Thank you very much for the constructive feedback from you and your reviewers. We have addressed all the reviewers' comments. Please find a point-by-point reply below.

Kind regards

Ayenew

**Response to Referee One**

The manuscript titled "Unveiling Hydrological Dynamics in Data-Scarce Regions: A Comprehensive Integrated Approach" authored by Ayalew et al., is a compelling work that is both well-structured and comprehensive. It provides a detailed analysis of hydrological changes in the Rift Valley lakes region, Ethiopia, presenting valuable and transferable methodological approaches in the context of regionalization, specifically in donor and pseudo watershed categorization. Additionally, the numerical analysis resulted in a valuable database for the study region.

However, I have two points that require clarification:

In reference to the SWAT+ modelling approach, the authors used the variable storage routing method for river routing. Could the authors provide justification for the choice of the variable storage routing method?

***Response:*** *There are two methods for channel routing currently implemented in SWAT+. These are Muskingum and Variable Storage Routing (VSR) methods.*

*The VSR is the standard method and has the following advantages*

1. *flexibility: it allows for variable storage coefficients for each reach, providing a more flexible representation of channel routing behavior*

2. *solving: it involves a set of differential equations that describe the dynamics of flow and storage in each channel reach. It introduces a diffusion wave approach to better represent the kinematics of wave propagation in a channel.*

3. *Representation: It considers the storage-discharge relationship for each reach, allowing for a better representation of channel routing processes.*

While the study effectively highlights substantial hydrologic changes in the region, the authors attribute these changes to climate change and human impacts. It would be beneficial for the authors

to elaborate on which driver—climate change or human impacts—had a more pronounced effect on altering the hydrologic regime in the study area. These clarifications would enhance the understanding of the methodology and findings presented in the manuscript.

***Response:*** *We appreciate the valuable feedback. We will incorporate this clarification into the discussion section of the paper: Our previous work has particularly targeted the attribution of Land Use and Land Cover (LULC) and climate change to the changes in hydrology in the study area (Ayalew et al. 2023). The findings indicate that the influence of human activities exerted a more significant impact compared to climate change. The hydrological regime of the lakes is affected by human-induced factors (mainly abstraction, urbanization, and deforestation) associated with rapid population growth. In the Northern part of the study area high water abstraction for irrigation and industry was the main driving factor for the changing hydrology. However, in the Southern part of the study area, high runoff and sedimentation associated with high deforestation are the main driving factors for the hydrological alterations.*

---

## Author Comment (AC2)

**Response to Anonymous Referees**

**Dear Editor Marleen de Ruiter ,**

Thank you very much for the constructive feedback from you and your reviewers. We have addressed all the reviewers' comments. Please find a point-by-point reply below.

Kind regards

Ayenew

**Response to Second Referee**

The authors aim to characterize the hydrological changes in two lakes in Ethiopia through a comprehensive evaluation employing a hydrological model (SWAT+) and bathymetry analysis. The study categorizes the time period into pre-impacted, impacted, and post-impacted phases to elucidate the observed changes. Results indicate a depletion of lake storage during the impacted period for both lakes, with varying recovery patterns—one lake exhibits increased storage compared to the pre-impacted period, while the other experiences a decline. The identified changes are attributed to factors such as climate change, land use alterations, and human water extraction.

The manuscript is well-crafted, particularly in the conclusion section; however, some sections, notably the results, require clarification. Below are suggested improvements:

**Title:** The title lacks a description of the study area. Given the regional focus, incorporating the study area in the title is recommended.

*Response: We appreciate the valuable feedback and the title has modified as per the comment*

***Unveiling Hydrological Dynamics in Data-Scarce Regions: The Ethiopian Rift Valley Lakes Basin***

**Abstract:**

P1 L20-21, 26: Providing %age values would be better.

*Response: Thank you for the valuable comment the value has changed to % age a depth decreased by 37.21%. The water level in Chamo Lake increased by 31%, with sedimentation accounting for a 4.23% increase.*

P1 L23-24: very small changes in ET compared to inflow changes.

*Response: Thank you very much for bringing this to our attention. It is a typo but increased by up to 4.5% in Chamo Lake.*

The introduction gives a good description of the study area, but it can be condensed. The contribution/novelty of the manuscript needs to be highlighted. Although the objectives are mentioned at the end of the introduction section, the research questions should be clearly articulated.

*Response:* *The introduction will be shortened and the contribution of the study will be highlighted and the research questions will be articulated as well. Thank you for the feedback.*

**Methods**:

The method section is well written. There are a few queries/suggestions though.

Study area: Is the 1st paragraph really needed?

*Response: The intention was to convey that the study area is situated within the Great Rift Valley and to provide its specific location. We acknowledge the feedback, and the 1*st *paragraph will be revised and shortened.*

The Ethiopian Rift Valley Lakes Basin is located between 36° and 40°E and 4° and 9°N. It extends from the Afar depression southwards to Kenya across the broad basins of Abijata-Ziway, Abaya-Chamo, and Segen (Fig. 1). It is one of the most important basins in Ethiopia and occupies an area of 55,050 km². It is characterized by diverse landscape features and climate conditions. It has a complex hydrological system and encompasses numerous lakes, springs, wetlands, and rivers. Given its distinctive attributes, it lends itself well to the development of a globally applicable methodology.

P6 L1: How is the slope % calculated?

*Response: The slope (%) is calculated as:*

*(Change in Elevation/ distance) *100*

$$Slope(\%) = \frac{Alt2 - Alt1}{x} * 100$$

*The study area is characterized by rugged topography and the elevation varies within in small distance.*

P6 L21: Why was SRTM DEM used despite the availability of better DEM data such as FABDEM and MERIT DEM?

*The SRTM DEM is a reliable and commonly used source of input data for hydrologic modeling. In our study, it is used for delineating the catchment and deriving the stream network as well as for the calculation of slopes that are then classified into slope class units. We believe that the finer (30 m) resolution of the SRTM DEM is an advantage in comparison to the coarser MERIT DEM. We agree with the reviewer that FABDEM would probably also be a good choice, but that a change from SRTM to FABDEM would not have major effects on the results of this study.*

P6 L21: Electricity?

**Response:** *Yes, it is Electricity. Thank you very much!*

P7, L22-24: I like the idea of dividing the durations into high-pulse and low-pulse periods. However, it was quite difficult for me to understand the pre-impacted, impacted, and post-impacted period. Kindly define these properly.

**Response:** *The periods before, during, and after the impact were determined through change point analysis, as elaborated in 1$^{st}$ paragraph of this section. However, it was observed that using the terms "pre-impacted," "impacted," and "post-impacted" in this section introduced confusion. Consequently, these terms were deliberately omitted to enhance clarity.*

P7 L13-22 can be incorporated into Table 2.

**Response:** We believe that the explanation of the groups is helpful in the text as it goes beyond the list of parameters presented in the table.

P8, L5/ P7 L13-22: Group3 data cannot be always associated with flood and drought. Low flow and high flow doesn't necessarily mean its flooding or it's a drought period. Also kindly explain the uncertainty associated with the dates (likely in the discussion section). There can be several dates within ±5% of the minimum and maximum values. Also, please explain how pulse count and pulse duration were calculated.

**Response:** *We agree that Group 3 data may not consistently correlate with flood and drought occurrences, particularly in the presence of water constructions such as dams used for irrigation, hydropower, factories, etc. Therefore, we have collectively decided to refrain from employing the terms "floods" and "droughts."*

*The calculation of pulse count and pulse duration was carried out through the utilization of the Indicators of Hydrological Alteration (IHA). A threshold for both high and low pulses is*

*used. Pulse count represents the number of pulses exceeding the established threshold (here the median), while pulse duration indicates the duration of persistence from the start to the end of a high pulse concerning the threshold.*

*The uncertainty concerning pulse counts, determined by the threshold, stems from potential overcounting and undercounting of low and high pulses. This occurs due to the computation of the median for the data series, which includes both flood and drought years. The presence of flood and drought years in the data series has the potential to influence the median value, consequently affecting the accuracy of the counted number of high and low pulses. Another threshold definition or other climate data will lead to different results.*

**Results**

Figures 4 and 5: kindly mention at least one matrix in the figure even though it is provided in Table 4.

***Response:*** *We are not sure if we get the comment right, but we can add the KGE values to figures 4 and 5.*

P20: The analysis is good. Kindly mention which data is modeled and which is observed. Also, please validate the modeled data wherever possible. One major concern I have here is that the enclosure term is quite large for both lakes.

***Response:*** *The data for ungauged river inflow and evapotranspiration are simulated and unfortunately, there is no available data to verify these results. To validate the simulated data, we developed the SWAT+ model for gauged catchments, also known as donor catchments. We tested these models using other gauged catchments, referred to as pseudo catchments. The enclosure term in this case is the water balance on the lake, which is determined by the difference between the input and output. The magnitude of this term depends on the amount of loss and gain.*

P21 L3 L9: Figure 8?

***Response:*** *Yes, it is figure 8. Thank you for pointing that out.*

P21 L4: on what basis were the impact periods determined?

***Response:*** *We employed change point analysis to identify the breakpoints.*

P21 L9: can you please briefly describe the environmental flow analysis in the methods section?

*Response: Thank you very much for the feedback. We will include environmental flow analysis in the method section with RVA.*

P21 L12: please don't use the word "flood" alternatively for the high flow and extreme high flow. These may be extreme flows, but not necessarily flood events.

*Response: Thank you for your feedback. We will use "high flow" and "extreme high flow" instead of "flood".*

P22 L7-8: Please reframe the sentence.

*Response: Will be rephrased to: The FDC showed an increase in the Abaya Lake level across all flow regimes (low, middle, and high) and a decrease in the Abijata Lake level.*

P23 L12: Kindly provide a reference for association with deforestation.

*Response: .....deforestation (Ayalew et al., 2022, Dessie and Kleman, 2007, Garedew et al., 2009).*

P24: It is very important to discuss the causes and impact of these changes, which I find missing. This would significantly increase the quality of the manuscript.

*Response: Thank you very much it is a good point. We will include it in the summary and conclusion sections. Eg. the ecosystem services will be affected.*

P27 L1-2: which changes led to these differences between the two lakes?

*Response: The main difference between the two lakes is their level of abstraction. The upper lake (Abijata) has high abstraction rates, while the lower part of the lake (Chamo) has no abstraction. Deforestation is more severe in the Lower Lake Basin compared to the Upper Lake Basin. This leads to more surface runoff generation.*

P27 L5-8: why does the lake level increase for one and decrease for the other?

*Response: The main reason is their level of abstraction. The upper lake (Abijata) has high abstraction rates, while the lower part of the lake (Chamo) has no abstraction (see also answer to the previous comment).*

P30 L4-5: Here and elsewhere, please mention %age change.

*Response: We will compute the % change and write in () as follows:*

*Specifically, the lake has experienced an expansion, gaining an area of 30 km² (9.5% ), and its volume has increased by 7.8 x 10⁸ m³ (27.2%) over the same period. As the maximum depth increased by 4.4 m (31%) and the lakebed level increased by 0.6 m (4.5%) possible factors that could contribute to such changes in lake size and volume are increased runoff and sedimentation from rivers or streams that flow into the lake due to land use change, deforestation.*

P30: Isn't the figure a part of figure 12? Please mention the caption.

*Response: Yes, it is. We will modify the figure to show all three diagrams as a part of one figure.*

**Discussions**

P32 L2-20: Can you please make it concise? Particularly L2-13 doesn't hold much scientific value.

*Response: Thank you for the feedback. We will only discuss the results.*

P33 L3: a **decrease** of

*Response: Thank you for the feedback. The typo error is corrected*

Additionally, address uncertainties in the study within the discussion section.

*Response: Thank you for the feedback. The comment will be considered.*

**References**

AYALEW, A. D., WAGNER, P. D., SAHLU, D., FOHRER, N. J. E. M. & ASSESSMENT 2022. Land use change and climate dynamics in the Rift Valley Lake Basin, Ethiopia. 194**,** 791.

DESSIE, G. & KLEMAN, J. 2007. Pattern and magnitude of deforestation in the South Central Rift Valley Region of Ethiopia. *Mountain research development,* 27**,** 162-168.

GAREDEW, E., SANDEWALL, M., SÖDERBERG, U. & CAMPBELL, B. M. 2009. Land-use and land-cover dynamics in the central rift valley of Ethiopia. *Environmental management,* 44**,** 683-694.

---

## Author Response (AR2)

**Dear Editor Marleen de Ruiter ,**

Thank you very much for the constructive feedback from you and your reviewer again. We have addressed the reviewers' comments not addressed before.

Kind regards

Ayenew

**Response to Anonymous Referees two**

Figures 4 and 5: kindly mention at least one matrix in the figure even though it is provided in Table 4.

*Response:* As requested the separation of the Model Evaluation Statistics table for figures 4 and 5, we have made the adjustment and included in the paper. The modification is indicated by the use of yellow highlighting on pages 18 and 19. Your valuable comment increased the clarity of our paper.